# Epidemiological characteristics of respiratory viruses in children during the COVID-19 epidemic in Chengdu, China

Zheng-Xiang Gao,[1,2] Ya Wang,[1,2] Ling-Yi Yan,[1,2] Ting Liu,[1,2] Lei-Wen Peng[1,2]

**ABSTRACT** The purpose of this retrospective study was to analyze the prevalence of respiratory viruses among children with acute respiratory tract infections (ARTIs) during the coronavirus disease 2019 epidemic (1 March 2020–28 February 2022). This study investigated respiratory viral specimens from children with ARTIs. A total of 7,092 children (<14 years) with ARTIs were included in this study, with a boy-to-girl ratio of 1.43. The median age was 1 year and 5 months. The average age of the patients was 2.7 ± 3.1 years. Patients < 3 years of age were the main population with ARTIs (67%). The predominant viruses were respiratory syncytial virus (RSV) (10.1%) and influenza virus (A and B 9.7%) during the 2 years of the pandemic (1 March 2020–28 February 2022). The proportion of positive viral test results among patients with ARTIs < 6 years of age was higher than that among patients with ARTIs aged 6–14 years (17.3% vs. 5.7%, $P < 0.01$). RSV infections were more common among patients < 3 years of age ($P < 0.01$). Influenza A infections were more common among patients aged 3–6 years ($P < 0.01$). Influenza B infections were more common among patients aged 6–14 years ($P < 0.01$). The positive proportion among boys was higher than that among girls (14.4% vs. 8.6%, $P < 0.01$). Peak virus infections occurred in the autumn and winter seasons, and the lowest activity level occurred in the spring and summer of the 2 years. Compared with that before the epidemic, the number of samples and positive proportion of respiratory viruses decreased significantly.

**IMPORTANCE** During the coronavirus disease 2019 epidemic, the Chinese government launched and used a series of nonpharmaceutical interventions (NPIs), including banning social gatherings, wearing face masks, home isolation, and maintaining hand hygiene, to control the disease spread. Whether and how NPIs influence other respiratory viruses in children remain unclear. In this article, we analyzed relative data and found that the number of samples and positive proportion of respiratory viruses decreased significantly compared with that before the epidemic. Clinicians and public health policymakers should pay attention to changes in the epidemic trends and types of respiratory viruses and maintain monitoring of respiratory-related viruses to avoid possible abnormal rebounds and epidemic outbreaks of these viruses.

**KEYWORDS** epidemiology, COVID-19, respiratory infection, virus

Acute respiratory tract infections (ARTIs) are the most common disease in children and adults (1, 2). ARTIs can occur regardless of age, sex, season, and region and are the main cause of infectious disease morbidity and mortality in China and worldwide (1, 2). Pathogens, including bacteria, mycoplasma, chlamydia, fungi and parasites, and viruses (such as influenza viruses, respiratory viruses, and rhinoviruses), have been identified as major causes of ARTIs in children (3–5). Studies have shown that acute viral respiratory tract infection is the leading cause of hospitalization among infants and children in developed countries (6) and is the leading cause of death in developing

Address correspondence to Lei-Wen Peng, lwpeng99@163.com.

The authors declare no conflict of interest.

See the funding table on p. 9.

countries (7, 8). At present, respiratory tract viruses, including human parainfluenza virus, respiratory syncytial virus (RSV), rhinovirus, coronavirus, adenovirus (ADV), and influenza virus, are the most common pathogens associated with ARTIs in children (9). Relevant studies have shown that controlling the spread of respiratory viruses is conducive to the growth and development of children. Since the immune and respiratory systems of infants and preschool children are not yet mature, early viral respiratory tract infections may have an adverse effect on lung development and increase the risk of incident asthma (10). ARTIs caused by viruses are the major cause of death and illness among children under 5 years of age (11). Therefore, the prevalence of respiratory viruses that cause ARTIs in children deserves our attention and continuous monitoring.

Coronavirus disease 2019 (COVID-19) is caused by severe acute respiratory syndrome coronavirus 2 (SARS-CoV-2). Since December 2019, COVID-19 has spread rapidly and subsequently become a pandemic. The World Health Organization declared COVID-19 a public health emergency of international concern on 30 January 2020. To control the spread of SARS-CoV-2, a series of nonpharmaceutical interventions (NPIs) were rapidly launched and used to control the spread of the disease worldwide. The categories of NPIs included banning social gatherings, wearing face masks, home isolation, and maintaining hand hygiene, which played an important role in controlling the spread of SARS-CoV-2 (12). By the end of March 2020, China had successfully controlled the first wave of the COVID-19 epidemic peak (13). In addition, studies have shown that NPIs affect the epidemic trends and transmission patterns of infectious diseases transmitted by air or feces, such as the common cold, gastroenteritis, bronchiolitis, and acute otitis media (14).

Our research group collected relevant data and analyzed the epidemiological characteristics of ARTIs caused by respiratory viruses in children before the outbreak of COVID-19 (10). At present, few studies have shown the epidemiology of respiratory viruses in children during the COVID-19 epidemic, and the incidence, age of onset, and seasonal changes in respiratory viruses in children remain unclear. Therefore, we retrospectively analyzed respiratory virus test data among children with ARTIs during the COVID-19 pandemic. Through the analysis of the positive proportion of seven respiratory viruses, common virus types, epidemic season, and age of onset, we aimed to clarify the impact of NPIs on non-COVID respiratory viruses among children during the COVID-19 pandemic. This could prevent antibiotic abuse and guide the effective treatment and prevention of ARTIs.

## MATERIALS AND METHODS

### Study population

To continue our previous research, we chose and retrospectively analyzed children with suspected ARTIs who visited pediatricians at West China Second University Hospital of Sichuan University from 1 March 2020 to 28 February 2022. The West China Second University Hospital of Sichuan University is located in Chengdu Sichuan province. The hospital was one of the first national top tertiary hospitals and serves as a medical center for women and children in southwestern China. It plays an important role in medical services, education, research, disease prevention, and healthcare. The hospital has 24 clinical departments/divisions and 7 medical supporting departments. As a referral center in Southwest China, we provide medical services to women and children with critical diseases. In 2022, we had 3.54 million outpatient and emergency visits, discharged 90,000 patients, performed 125,000 operations, and delivered 22,000 infants. Chengdu's resident population was 21.26 million, and the number of 0–14-year-old children was 2.78 million at the end of 2022. The inclusion criteria were as follows: (i) patients younger than 14 years of age (as in China, the children visiting pediatric clinics are mainly aged 0–14 years.); (ii) patients with at least the following two signs or symptoms: fever (>38.0℃), sore throat, cough, hoarseness, runny nose, nasal congestion/pharyngeal purulent exudate, increased respiratory rate, or moist rales; and (iii)

ARTIs mentioned in the outpatient or discharge diagnosis. The exclusion criterion was the lack of information on ARTIs in the top three outpatient or discharge diagnoses. All patients were examined and diagnosed by clinicians.

## Sample evaluation

Nasopharyngeal swabs were collected from patients ($n = 7092$) in the outpatient or inpatient ward and sent to the hospital laboratory for testing. After receiving the nasopharyngeal swabs, the laboratory technicians immediately processed and tested the samples. Seven respiratory viruses, including RSV; influenza A and B; parainfluenza viruses (PIV) I, II, and III; and ADV, were detected by a direct immunofluorescence assay (Diagnostic Hybrids Inc., Athens, OH, USA) using nasopharyngeal swabs. The direct immunofluorescence assay is relatively simple and inexpensive and is widely used in clinical respiratory virus testing. Quality control will be performed in each test to ensure that the results are reliable. At the same time, if the sample test results were suspicious, they needed to be verified by PCR before inclusion in this study. The direct immunofluorescence assay kit did not contain metapneumovirus; to ensure the consistency of the results, there was no metapneumovirus testing. The clinician tested for metapneumovirus by PCR.

## Data collection

The laboratory test results and patient demographic data (name, sex, age, clinical diagnosis, and sampling time) were extracted from the laboratory information system (LIS) of West China Second Hospital of Sichuan University. The study protocol was approved by the Ethics Committee of West China Second University Hospital, Sichuan University.

## Statistical analysis

The data were analyzed by SPSS Statistics 19 (SPSS Inc., Chicago, IL). All variables, including age and sex, were tested for a normal distribution. Age was compared by the Mann−Whitney $U$ test. Due to the unequal population size and data differences, the $\chi^2$ test or Fisher's exact test was used to analyze the association between age and sex with positivity for respiratory viruses. $P < 0.05$ was considered statistically significant.

## RESULTS

From 1 March 2020 to 28 February 2022, 7,092 patients who came to the hospital and were diagnosed with ARTIs were included in this study. A total of 638 patients were excluded. The sociodemographic variables associated with all samples are summarized in Table 1. Regarding the age distribution of patients, patients aged 6 months to 3 years were the main population with ARTIs (35.3%), followed by patients aged 28 days to 6 months (20.3%). The median age was 1 year and 5 months (range: 0 days to 14 years). The average age of the patients was 2.7 ± 3.1 years. Patients < 3 years of age were the main population with ARTIs (67%). There were significantly more boys than girls (58.9% vs. 41.1%, $P < 0.01$). The number of inpatients was significantly higher than that of outpatients (61.2% vs. 38.8%, $P < 0.01$). However, the positive proportion of virus detection among inpatients was significantly lower than that among outpatients (3.8% vs. 19.3%, $P < 0.01$).

Regarding the prevalence of respiratory viruses among children with ARTIs, of the 7,092 patients, 1,635 patients (23.1%) had respiratory viruses. The positive proportions of influenza A, influenza B, PIV I, PIV II, PIV III, ADV, and RSV were 4.1%, 5.6%, 0.7%, 0.1%, 1.1%, 1.3%, and 10.1%, respectively (Table 1). The predominant viruses were RSV (10.1%, $P < 0.01$) and influenza (A and B 9.7%, $P < 0.01$) during the study period, followed by PIV (I, II, and III, 1.9%) and ADV (1.3%). Among patients of different ages, the proportion of positive viral test results among patients < 6 years of age was significantly higher than that among patients aged 6–14 years (17.3% vs. 5.7%, $P < 0.01$). RSV infections were more common among patients < 3 years of age (9.4%, 667/7,092, $P < 0.001$). The positive

**TABLE 1** Characteristics of included children

| Characteristic | Age | | | | | Sex | | Category | | Total |
|---|---|---|---|---|---|---|---|---|---|---|
| | 0–28 d | 28 d–6 mo | 6 mo–3 yr | 3–6 yr | 6–14 yr | Boy | Girl | Outpatient | Inpatient | |
| Samples (% of total samples received) | | | | | | | | | | |
| Total samples received | 808 (11.4) | 1,440 (20.3) | 2,505 (35.3) | 1,232 (17.4) | 1,107 (15.6) | 4,175 (58.9) | 2,917 (41.1) | 2,749 (38.8) | 4,343 (61.2) | 7,092 (100) |
| Positive samples | 34 (0.5) | 364 (5.1) | 516 (7.3) | 314 (4.4) | 407 (5.7) | 1,024 (14.4) | 611 (8.6) | 1,368 (19.3) | 267 (3.8) | 1,635 (23.1) |
| Respiratory virus identified (% of total positive samples based on age/sex) | | | | | | | | | | |
| Influenza A | 0 (0.0) | 2 (0.5) | 36 (7.0) | 152 (48.4) | 104 (25.6) | 202 (19.7) | 92 (15.1) | 282 (20.6) | 12 (4.5) | 294 (4.1) |
| Influenza B | 0 (0.0) | 4 (1.1) | 50 (9.7) | 65 (20.7) | 277 (68.1) | 230 (22.5) | 166 (27.2) | 389 (28.4) | 7 (2.6) | 396 (5.6) |
| Parainfluenza virus I | 2 (5.9) | 1 (0.3) | 29 (5.6) | 16 (5.1) | 0 (0.0) | 24 (2.3) | 24 (3.9) | 34 (2.5) | 14 (5.2) | 48 (0.7) |
| Parainfluenza virus II | 0 (0.0) | 0 (0.0) | 2 (0.4) | 2 (0.6) | 2 (0.5) | 2 (0.2) | 4 (0.7) | 5 (0.4) | 1 (0.4) | 6 (0.1) |
| Parainfluenza virus III | 1 (2.9) | 21 (5.8) | 45 (8.7) | 10 (3.2) | 3 (0.7) | 50 (4.9) | 30 (4.9) | 46 (3.4) | 34 (12.7) | 80 (1.1) |
| Adenovirus | 0 (0.0) | 15 (4.1) | 39 (7.6) | 23 (7.3) | 17 (4.2) | 62 (6.1) | 32 (5.2) | 77 (5.6) | 17 (6.4) | 94 (1.3) |
| Respiratory syncytial virus | 31 (91.2) | 321 (88.2) | 315 (61.0) | 46 (14.6) | 4 (1.0) | 454 (44.3) | 263 (43.0) | 535 (39.1) | 182 (68.2) | 717 (10.1) |

proportion of influenza A among patients aged 3–6 years was significantly higher than that of other viruses ($P < 0.01$). Influenza B infections were more common in patients aged 6–14 years ($P < 0.01$) (Fig. S1). The positive proportion among boys was higher than that among girls (14.4% vs. 8.6%, $P < 0.01$). The seasonal distribution of the sample positive proportion during the study period is shown in Fig. 1. The positive proportions of boys, girls, and total in the autumn and winter seasons were significantly higher than those in the spring and summer seasons during the study period. In the first year, the positive proportion of boys was highest in January (24.4%) and lowest in July (2.2%) (Fig. 1A). The positive proportion of girls was highest in January (11.5%) and lowest in May (1.4%) (Fig. 1B). The total positive proportion was highest in January (36.0%) and lowest in July (3.9%) in this year (Fig. 1C). In the second year, the positive proportion of boys was highest in October (44.1%) and lowest in June (5.0%) (Fig. 1A). The positive proportion of girls was highest in October (30.5%) and lowest in April to June (3.2%–3.3%) (Fig. 1B). The total positive proportion was highest in October (74.6%) and lowest in June (8.3%) in this year (Fig. 1C). Meanwhile, the positive proportion of boys was higher than that of girls.

The seasonal distributions of each respiratory virus in the first and second years are shown in Fig. 2 and 3, respectively. As shown in Fig. 2A, the peak infection period of influenza A and RSV occurred in the autumn and winter seasons (from October 2020 to February 2021), and the lowest activity level occurred in spring and summer (from March to August). The positivity of other viruses was low, but their peak infection periods were also in the autumn and winter seasons (from September to December). The number of positive samples in the first year was 643, and the predominant virus was RSV, accounting for 53.5% of the positive samples, followed by influenza A, accounting for 23.3% of the positive samples. The third major viruses were ADV (8.2%) and PIV III (6.7%). The positive proportion of the other three viruses was low (Fig. 2B). Among the positive samples, 64.9% and 35.2% were isolated from boys and girls, respectively (Fig. 2C). The main positive proportion of virus detection among boys was RSV and influenza A, and the main positive proportion of virus detection among girls was RSV and influenza A (Fig. 2D). A total of 77.9% of the positive samples were from outpatients and 22.0% were from inpatients (Fig. 2E). The age distribution of positive patients was as follows: most patients were aged 6 months to 3 years ($n = 270$), followed by those aged 28 days to 6 months ($n = 151$) and 3–6 years ($n = 140$) (Fig. 2F).

The number of positive samples was 992 in the second year, and the number of positive samples for influenza A and RSV was similar to that in the first year. However, the number of positive cases of influenza B increased significantly, and there was an obvious epidemic peak in the autumn and winter seasons (Fig. 3A). The predominant viruses in 2021 were influenza B and RSV, accounting for 38.6% and 37.6%, respectively. The third most predominant virus was influenza A, with a 14.5% positivity rate, and the positive

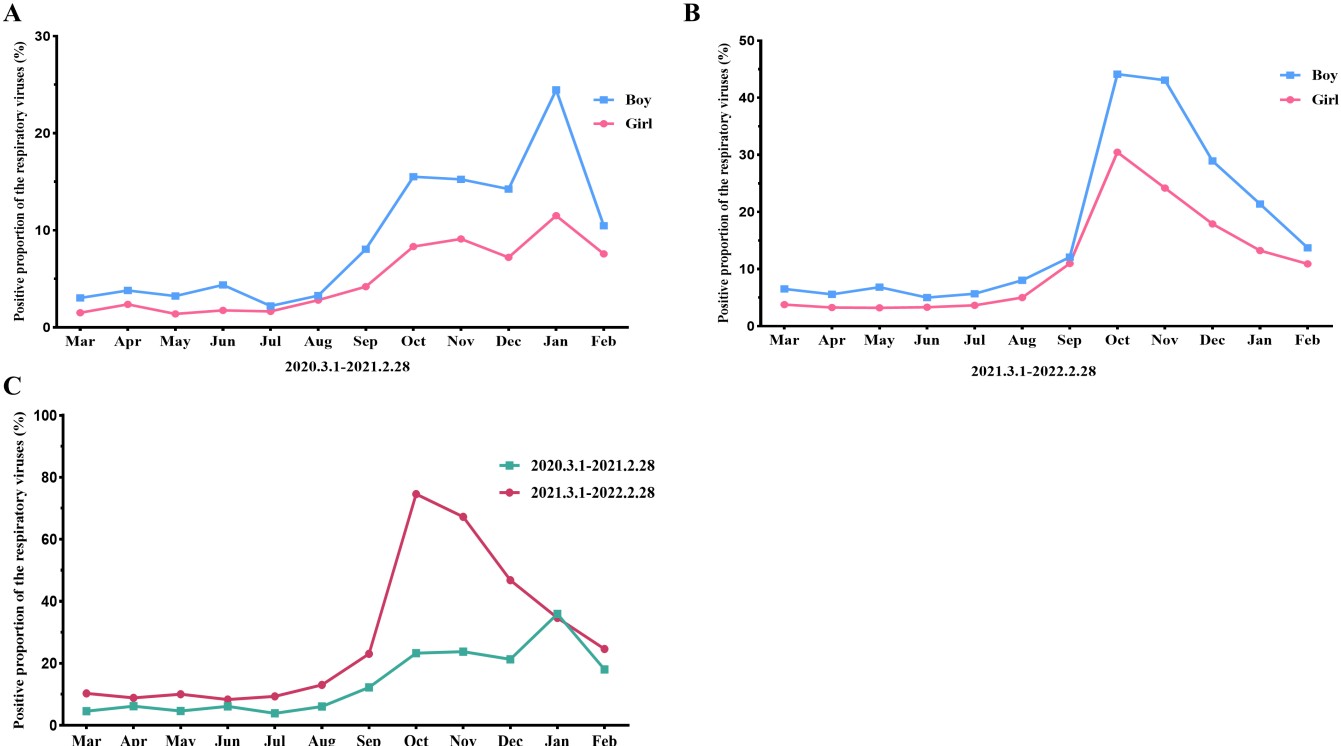

**FIG 1** Monthly activity patterns of the positive proportion of respiratory viruses. (A) Monthly activity patterns of the positive proportion of respiratory viruses from March 1, 2020, to February 28, 2021. (B) Monthly activity patterns of the positive proportion of respiratory viruses from 1 March 2021 to 28 February 2022. (C) The different monthly activity patterns of the positive proportion of respiratory viruses between the 2 years.

proportion of the other four viruses was low (Fig. 3B). Among all positive samples, 61.2% came from boys and 38.9% came from girls (Fig. 3C). The main positive proportion of virus detection among boys was RSV and influenza A and B, and the main positive proportion of virus detection among girls was RSV and influenza A and B (Fig. 3D). A total of 87.4% were outpatients and 12.6% were inpatients (Fig. 3E). Compared with the previous year, there was no statistically significant difference in the proportion of boys and girls or inpatients and outpatients ($P > 0.05$). The age distribution of positive patients was as follows: most patients were 6–14 years old ($n = 340$), followed by 6 months to 3 years old ($n = 246$) and 28 days to 6 months old ($n = 213$) (Fig. 3F).

## DISCUSSION

ARTIs account for a large proportion of infectious diseases in children, and respiratory viruses are the most common cause of ARTIs in children and infants (15, 16). Although the clinical characteristics of children and infants infected with different kinds of respiratory viruses are different, their transmission routes are nearly the same, mainly through droplets and aerosols and direct or indirect contact with infected secretions. This study focused on the prevalence of respiratory virus infections in Southwest China during the COVID-19 pandemic, aiming to investigate the impact of NPIs on the prevalence of respiratory viruses among children and infants.

In this study, we first counted the number of children with ARTIs who were tested for viral antigens during the COVID-19 pandemic. The total number of children tested was 7,092 from March 2020 to March 2022. Compared with the related data published by our research group from March 2018 to March 2020, the number of patients decreased substantially (7,092 vs. 11,813), and the positive proportion also decreased to 3/5 of that before the outbreak of COVID-19 (10). Studies have shown that the number of pediatric outpatients and inpatients in many countries decreased substantially in 2020, when the COVID-19 pandemic began (17–19). A study showed that the number of children

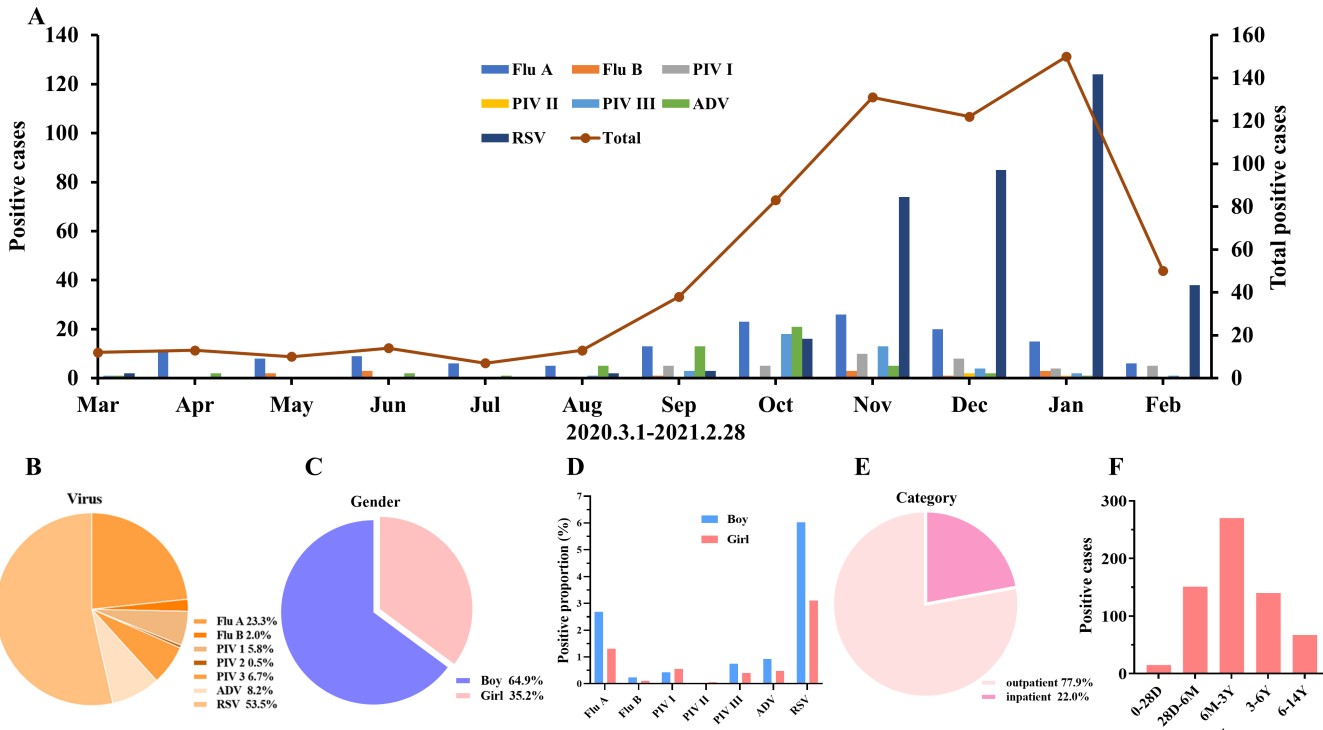

**FIG 2** The epidemic trend and demographic characteristics of respiratory viruses in the first year. (A) Monthly activity patterns of viruses. (B) The proportions of different viruses in the first year. (C) The proportions of different sexes among positive children. (D) The proportions of different viruses among positive children. (E, F) The proportions of different categories and ages of positive children. Flu A, influenza A; Flu B, influenza B; PIV I, parainfluenza virus I; PIV II, parainfluenza virus II; PIV III, parainfluenza virus III; ADV, adenovirus; RSV, respiratory syncytial virus; D, day; M, month; Y, year.

hospitalized for infectious diseases decreased by 37%, among whom the number of children with respiratory infections decreased by 33% (20). Similar reductions were observed in this study, with decreases of 31% for inpatients. At the same time, the study found that the relevant reasons may be related to the adoption of serious intervention measures, i.e., social distancing and especially school closures, leading to a sharp reduction in contact between children, which may have reduced the risk of ARTI transmission. On the other hand, parents' concerns about sick children and their seeking of timely clinical care may not play an important role (20). Moreover, the Chinese government implemented similar public preventive measures after the outbreak of COVID-19. Our results showed that the number and positive proportion of respiratory viruses among children of all ages decreased significantly. This finding indicates that NPIs to control SARS-CoV-2 transmission were also effective in reducing the transmission of respiratory viruses among children during the COVID-19 pandemic.

During the study period, the number of boys infected with ARTIs and the positive proportion of virus detection were significantly higher than those of girls, which is consistent with previous studies, and it is reported that boys may be more susceptible to ARTIs than girls (10, 21). The study revealed that the genetic susceptibility of children to respiratory diseases is related to the type of virus and specific respiratory diseases of patients. There are few studies indicating that the genetic susceptibility of children to respiratory infectious diseases is related to sex (22–24). Meanwhile, in the gender composition of children in Chengdu, boys accounted for 50.26%, girls accounted for 49.74%, and the proportion of boys and girls was basically the same. Therefore, we did not normalize the ratio of virally infected boys vs. girls to an uninfected population.

Children under 3 years of age were the main population with ARTIs (67%). The possible reason is the waning of maternal antibodies after 6 months and the fact that the immune system gradually matures after 3 years of age. At this age stage, children lack

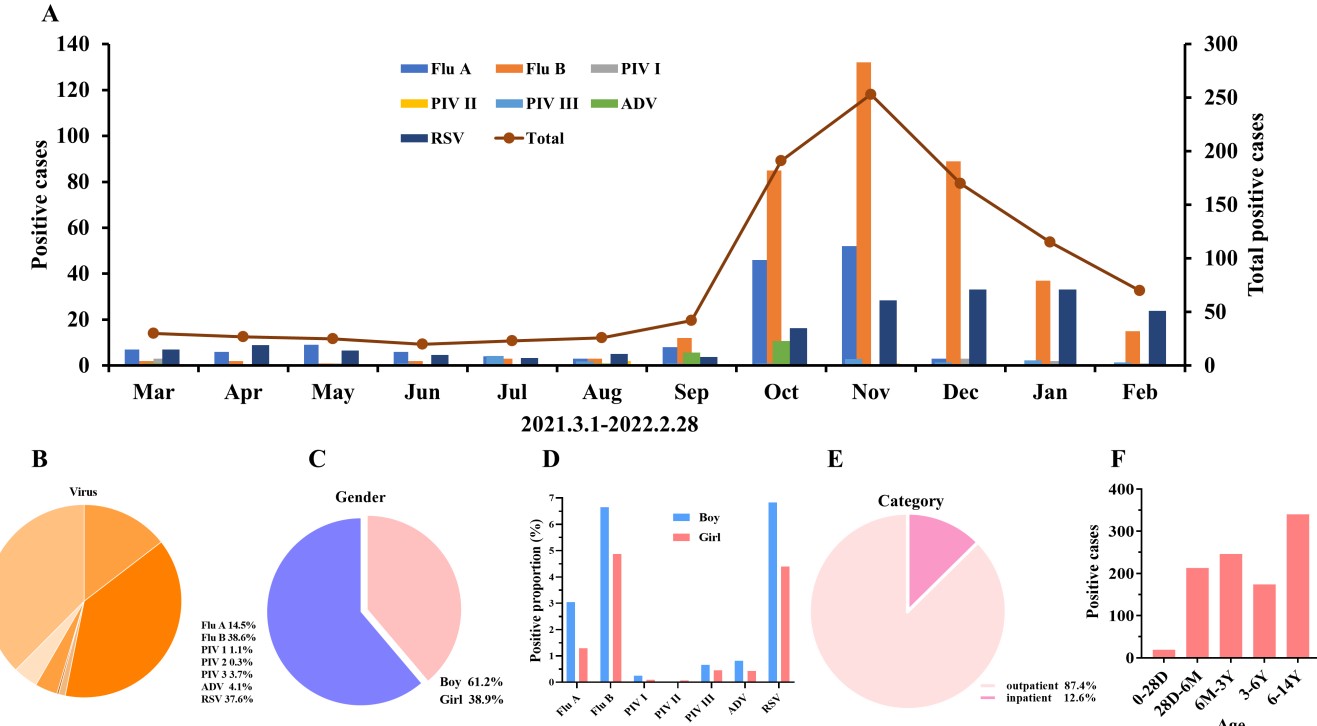

**FIG 3** The epidemic trend and demographic characteristics of respiratory viruses in the second year. (A) Monthly activity patterns of viruses. (B) The proportions of different viruses in the first year. (C) The proportions of different sexes in positive children. (D) The proportions of different viruses in positive children. (E, F) The proportions of different categories and ages of positive children. Flu A, influenza A; Flu B, influenza B; PIV I, parainfluenza virus I; PIV II, parainfluenza virus II; PIV III, parainfluenza virus III; ADV, adenovirus; RSV, respiratory syncytial virus; D, day; M, month; Y, year.

antibodies against viral infections and are more easily infected by viruses (25). In this study, the number of inpatients was significantly higher than that of outpatients. The possible reason is that this testing requires a long time and outpatients hope to obtain the test results as soon as possible to visit the doctor, resulting in a smaller number of tests than those for inpatients. At the same time, the main positive proportion of virus detection among outpatients involved RSV, influenza A and B, and the main positive proportion of virus detection among inpatients involved RSV, which is consistent with the relevant research reported (26, 27).

Our previous research revealed that children under 5 years of age were mainly infected with RSV and influenza A, and children aged 5–10 years were mainly infected with influenza A (10). During the pandemic, we found that children under 3 years of age were mainly infected with RSV, children aged 3–6 years were mainly infected with influenza A, and children aged 6–14 years were mainly infected with influenza A. Relevant studies have also shown that the population susceptible to RSV is mainly infants and young children and the detection rate of RSV decreased with age, owing to the maturation of the immune system (28, 29). According to statistics, RSV causes at least 3.4 million hospitalizations and between 66,000 and 199,000 deaths among young children under 5 years of age every year (30). Meanwhile, the influenza virus prevalence increased with age, which was similar to previous reports (31, 32). Based on the results of this study, it is possible that there was no significant change in the population susceptible to viruses after the COVID-19 pandemic. Our previous data showed that influenza A was the most common virus in this area before the COVID-19 pandemic, followed by RSV and influenza B (10). In this study, the most common virus detected in children with ARTIs was RSV, followed by influenza A and B 2 years after the COVID-19 pandemic. This may have been due to NPIs applied during the COVID-19 pandemic, during which student learning changed from traditional classroom teaching in schools to online learning at home. School-age children are susceptible to influenza infection, and

decentralized teaching reduces the risk of this infection. Combined with the increased number of RSV infections, influenza A and B infections may be mainly transmitted in schools, while RSV may be mainly transmitted within the family or community. Additionally, influenza virus vaccination may not be a convincing factor. Under the influence of COVID-19, the willingness to receive influenza vaccines has increased to avoid double infection with influenza viruses and SARS-CoV-2 (33). Even so, compared with other countries, China's influenza vaccination coverage is very low, far from the herd immunity threshold (34). The vaccination of COVID-19 vaccine in Chinese children began in October 2021, and the first round of vaccination was completed at the end of December 2021. We analyzed the rates of positivity for respiratory viruses in January 2022–February 2022. Compared with the positive proportion for respiratory viruses from January to February 2021, there was no significant change in 2022. The observation time after the new coronavirus vaccination in children was only 2 months, and a longer time is needed in future studies.

The positive proportion of viral infections increased significantly between September 2021 and January 2022. Further analysis showed that the positive proportion of influenza B was significantly higher than that of other viruses, thus increasing the total positive proportion of the virus. At the same time, the data released by the National Influenza Center of China showed that the number of cases of influenza B infection increased substantially during this period (35). Therefore, we speculate that the increased positive proportion of influenza B may be related to the influenza B epidemic. After excluding cases of influenza B, the monthly trend of the positive proportion of respiratory virus detection during the study period was generally consistent. Although the positive proportions in the autumn and winter seasons were higher than those in the spring and summer seasons, the degree of change in the positive proportion by seasonality was significantly reduced compared with that before the COVID-19 epidemic (10). Related studies have also shown that seasonal changes in the positive proportion of respiratory viruses were due to NPIs preventing SARS-CoV-2 transmission (36, 37). The number of RSV cases increased considerably after New Zealand partially relaxed its strict border lockdown policy in April 2021 to five times the average peak level of 2015–2019 (38). Meanwhile, the impact of non-NPIs may have had an effect on reducing the rate of respiratory virus infection in children. Parents are not willing to take their sick children to the hospital during the COVID-19 pandemic. They are worried about nosocomial infections, unwilling to be isolated from their children, or reluctant to burden hospitals during the COVID-19 pandemic. Consequently, they may choose to delay treatment or seek medical help online. On the one hand, this reduces the spread of respiratory viruses; on the other hand, it also reduces the number of respiratory viruses detected (39). At the same time, due to isolation, less travel reduces air pollution, parents have more time to take care of sick children at home, and children may recover faster, thus indirectly reducing the probability of respiratory virus infection (40). The introduction of COVID-19 containment policies and public awareness campaigns may have reduced the transmission of *Streptococcus pneumoniae*, *Haemophilus influenzae*, and *Neisseria meningitidis*, thereby greatly reducing life-threatening invasive diseases in many countries worldwide (41). Studies have revealed that the number of admissions for respiratory infections in children caused mainly by bacteria has decreased significantly. The number of admissions for pneumonia decreased by 60%, that for otitis media decreased by 74%, and that for tonsillitis decreased by 66% after the outbreak of the epidemic. The number of bacterial and viral coinfections also decreased (42).

This study has some limitations. First, this was a single-center study. We only collected data on children who visited our hospital for respiratory virus testing. These patients were more likely to be seriously affected, which may have led to selection bias. In the future, multicenter research trials and larger patient cohorts will provide a better reference and support for the control, prevention, and treatment of respiratory virus infections. Second, this was a retrospective analysis. We included only a limited number of factors and investigated some viral respiratory antigens through the final serological

positive proportion of the virus; therefore, further epidemiological monitoring of more pathogens through molecular assays and their epidemic subtypes is needed to reveal more detailed epidemiological information. Third, the comorbidities of patients were not analyzed in this study; we will focus on this topic in future investigations.

In this study, we recruited more than 7,000 patients with ARTIs and analyzed the prevalence of respiratory viruses during the COVID-19 pandemic in Chengdu, China. Compared with that before the epidemic, the number of samples and positive proportion of respiratory viruses decreased significantly (40% and 47%, respectively). Clinicians and public health policymakers should pay attention to changes in the epidemic trends and types of respiratory viruses and maintain monitoring of respiratory-related viruses to avoid possible abnormal rebounds and epidemic outbreaks of these viruses.

## ACKNOWLEDGMENTS

This work was supported by the Foundation (No. 21H1219 and No. 22H1407) and the Science & Technology Department of Sichuan Province (No. 2023YFS0186). The funders played no role in the study design, data collection and analyses, decision to publish, or manuscript preparation.

## AUTHOR AFFILIATIONS

[1]Department of Laboratory Medicine, West China Second University Hospital, Sichuan University, Chengdu, Sichuan, China
[2]Key Laboratory of Birth Defects and Related Diseases of Women and Children, Sichuan University, Ministry of Education, Chengdu, Sichuan, China

## AUTHOR ORCIDs

Zheng-Xiang Gao  http://orcid.org/0000-0003-4042-8960
Lei-Wen Peng  http://orcid.org/0000-0001-7800-7067

## FUNDING

| Funder | Grant(s) | Author(s) |
| --- | --- | --- |
| Science and Technology Department of Sichuan Province (SPDST) | No. 2023YFS0186 | Ting Liu |

## DATA AVAILABILITY

The raw data supporting the conclusions of this article will be made available by the authors, without undue reservation.

## ETHICS APPROVAL

The study protocol was approved by the Ethics Committee of West China Second University Hospital of Sichuan University.

## ADDITIONAL FILES

The following material is available online.

Supplemental Material

**Supplemental Figure S1 (Spectrum02614-23-s0001.docx).** The differences in age and the different viral infections. Flu A = influenza A; Flu B = influenza B; PIV I = parainfluenza virus I; PIV II = parainfluenza virus II; PIV III = parainfluenza virus III; ADV = adenovirus, RSV = respiratory syncytial virus, D = Day, M = Month, Y = Year.

## Open Peer Review

**PEER REVIEW HISTORY (review-history.pdf).** An accounting of the reviewer comments and feedback.

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
