## [Reviewer comments · Microbiology Spectrum]

Microbiology Spectrum

Epidemiological characteristics of respiratory viruses in children during the COVID-19 epidemic in Chengdu, China

Zheng-Xiang Gao, ya wang, Ling-Yi Yan, ting liu, and leiwen peng

Corresponding Author(s): leiwen peng, Sichuan University West China Second University Hospital

Review Timeline:

Submission Date:	June 22, 2023
Editorial Decision:	September 7, 2023
Revision Received:	October 6, 2023
Editorial Decision:	October 27, 2023
Revision Received:	October 29, 2023
Accepted:	October 31, 2023

Editor: Tulip Jhaveri

Reviewer(s): Disclosure of reviewer identity is with reference to reviewer comments included in decision letter(s). The following individuals involved in review of your submission have agreed to reveal their identity: Nazly Shafagati (Reviewer #2)

Transaction Report:

DOI: <https://doi.org/10.1128/spectrum.02614-23>

September 7, 2023

Ms. leiwen peng
Sichuan University West China Second University Hospital
#20, Section 3, South Ren Min Road, Chengdu, Sichuan
Chengdu, sichuan 610041
China

Re: Spectrum02614-23 (Epidemiological characteristics of respiratory viruses in children during the COVID-19 epidemic in Chengdu, China)

Dear Ms. leiwen peng:

Link Not Available

Sincerely,

Tulip Jhaveri

Journals Department
Reviewer comments:

Reviewer #1 (Comments for the Author):

Summary of Key Findings:

This retrospective study aimed to analyze the prevalence of respiratory viruses in children with acute respiratory tract infections (ARTIs) during the COVID-19 pandemic. The study included 7,092 children with ARTIs, with a higher proportion of boys and a median age of 1 year and 5 months. The main population affected were children under 3 years of age. The predominant viruses detected were respiratory syncytial virus (RSV) and influenza A and B. The positive rate of viral testing was higher in children under 6 years of age compared to those aged 6-14 years. RSV infections were more common among children under 3 years

old, while influenza A and B infections were more common in older children. Boys had a higher positive rate than girls. Virus infections peaked in the autumn and winter seasons and were lowest in spring and summer. The study found a significant decrease in the number of samples and positive rates of respiratory viruses compared to before the COVID-19 epidemic, possibly due to nonpharmaceutical interventions (NPIs) implemented during the pandemic.

Broadly, the authors demonstrate that the number of ARTIs decreased, however, the expected relative positivity proportions of ARTIs by species, age and seasonality remained consistent during the COVID-19 pandemic implying that NPIs impacted non-COVID-19 respiratory viruses in a somewhat predictable way that can be applied in future pandemics or during respiratory seasons.

Major Concerns:

- The lack of vaccination status (COVID-19, Flu, etc.) impacts the generalizability and interpretation of the results. The authors note this as a limitation and also note China's relatively low Flu vaccination rates.
- The impact of non-NPI pandemic-related factors is not discussed. I would advise a section of the discussion dedicated to addressing some of these factors, such as: reporting practices, variations in patterns of other infectious diseases/co-infections, changes in healthcare-seeking behavior, mistrust in science/healthcare, etc.

Minor Concerns:

Broad:

- Given the nature of a single center study, I would advise including details about the medical center and the patient population to give readers better context: geographic info, socioeconomic data, healthcare access, etc.
- Were comorbidities of patients analyzed? If so, I would include this data as it can impact ARTIs.
- Are there any available measures of adherence to NPIs as time went on in the 2-year study? As pandemic fatigue set in, were people more likely to not adhere therefore increasing ARTIs over the study period?

Abstract

- Define the 2 years of the pandemic as dates vary between countries
- Define the age of study 'children' even if typical <18 years old
- "Positive rate" should be changed to "Positive proportion" unless the data was analyzed over time.
- How were positive test proportions calculated if only children with ARTIs were included? If data is available on all tested children, meaning those who tested negative too, it should be presented, even if briefly, to give context.
- NPIs are one of many items that could have influenced any change in ARTIs during the pandemic. NPIs should be defined here so readers are clear this includes masking, distancing, etc.
- "The types of epidemic viruses [changed]". I do not see any context that backs up this claim in the conclusion. This should be made clearer or removed.

Introduction

- Line 42 needs a citation if not included in line 43.
- Line 52: for all patients or just children?
- Line 56: citation
- Line 68: Is the year 2020 accurate to the epidemic? Or is this intended for the first 'wave'? Peaks occurred later than 2020 in China.
- Lines 79-85: It's confusing to connect these two ideas: 1) NPIs likely impacted ARTIs during the COVID pandemic and 2) understanding the prevalence of respiratory viruses during the pandemic could provide early warning of possible epidemic respiratory diseases. Are we attempting to understand the impact of NPIs on non-COVID respiratory viruses or a baseline level of viral infection prevalence's?

Methods

- Line 88: Why were these specific dates chosen as bookends? Even if obvious, please add.
- Please add the size and basic description of the study hospital for context.
- Why was 14 years of age chosen?
o Throughout the manuscript the age inclusion criteria of <14 years of age should be emphasized as it is not a standard age cutoff.
- Were nasal swabs collected during the healthcare encounter patients were originally identified in or were they recruited and returned? This is critical considering the sensitivity of assays decreases as time from symptom onset increases.
- Were nasal swabs collected and tested for clinical purposes and done in the clinical microbiology lab or were these completed by research staff?
- How many patients were excluded?

Discussion

- Line 215: Is this finding consistent with previous studies or is it unique to this one?
- Line 218: The assumption of activity level differences between boys and girls should be removed unless data or relevant citations specific to the study area are added.

Reviewer #2 (Comments for the Author):

The paper is overall a very interesting perspective on how strict restrictions implemented to curb the spread of Covid-19 infection had an effect on other respiratory pathogens. Gao et al took a retrospective look at samples obtained from March 2020 to March 2022, several months prior to the restriction being lifted in China. The paper is a simple yet effective look into how viral respiratory trends lifted, including a shift from influenza A dominance to RSV dominance, decrease in cases at the centre where specimens were collected, and a remaining bias towards boys above the age of 3 having a higher likelihood for infection. The limitation of the study include and are not limited to being a single-centre study, samples only undergoing initial direct immunofluorescence assay testing and no PCR testing, not properly examining differences in viruses from inpatient vs outpatient patients, and the exclusion of findings for any co-infections.

Staff Comments:

Preparing Revision Guidelines

Please return the manuscript within 60 days; if you cannot complete the modification within this time period, please contact me. If you do not wish to modify the manuscript and prefer to submit it to another journal, please notify me of your decision immediately so that the manuscript may be formally withdrawn from consideration by Microbiology Spectrum.

Reviewer Comments for Gao et al

The paper is overall a very interesting perspective on how strict restrictions implemented to curb the spread of Covid-19 infection had an effect on other respiratory pathogens. Gao et al took a retrospective look at samples obtained from March 2020 to March 2022, several months prior to the restriction being lifted in China. The paper is a simple yet effective look into how viral respiratory trends lifted, including a shift from influenza A dominance to RSV dominance, decrease in cases at the centre where specimens were collected, and a remaining bias towards boys above the age of 3 having a higher likelihood for infection. The limitation of the study include and are not limited to being a single-centre study, samples only undergoing initial direct immunofluorescence assay testing and no PCR testing, not properly examining differences in viruses from inpatient vs outpatient patients, and the exclusion of findings for any co-infections.

There are some minor grammatical errors that should be addressed:

Line 53 - "since" instead of "because"

Line 58 – "deserve"

Line 82 – remove "us"

Other comments:

Abstract

Line 13 – study examined only viral respiratory specimens

Lines 14 and 19 – include the time frame in abstract

Line 25 – peak virus infections occurred in the autumn and winter seasons in both years examined?

Materials and Methods

Line 98 – Any PCR testing performed? Any repeat testing?

Results

Did the authors or clinicians look into coinfections?

Why did the clinicians not include metapneumovirus as part of their testing?

Line 136 – Suggestion to authors to provide a figure to show the differences in age and the different viral infections? Table 1 should be mentioned again.

Line 161 – authors mention the distribution of positive cases and viruses in boys versus girls during the pandemic. Please also mention distribution see prior to Covid pandemic.

Lines 132, 157 and 167 – The authors state the number of positive samples were 637 the first year and 992 the second year; these numbers do not match up to 1635 total positive samples reported

Discussion

The authors should elaborate more on the differences in viruses seen in outpatient vs inpatient samples.

Although significant, a change from 11813 samples precovid to 7092 samples reported during covid is not as dramatic due to the lockdowns. Was this also due to less testing done prior to covid?

Line 201 – state percentage of decrease for children with respiratory infections and compare this percentage with the children in this study

Lines 206-208 is unclear and difficult to understand. “May not have had as great an effect” is unclear.

Line 215 – is there also a genetic predisposition to boys having a higher rate of infections? Were the ratio of virally-infected boys vs girls normalized to an uninfected population?

Line 220 – Authors should provide the citation in their own words.

Line 224 – The paragraph can be reworded to better compare what was first previously seen and what is seen during the pandemic

Line 280 – Include % decrease

Figures

Figure 1 Include 2018-2020 data if possible. Can be broken up to three different figures and further broken down by boys, girls, and total.

Figures 2 and 3 Another figure can be included to further break down the different rates of infection in boys vs girls

Reviewer comments:

Reviewer #1 (Comments for the Author):

Summary of Key Findings:

This retrospective study aimed to analyze the prevalence of respiratory viruses in children with acute respiratory tract infections (ARTIs) during the COVID-19 pandemic. The study included 7,092 children with ARTIs, with a higher proportion of boys and a median age of 1 year and 5 months. The main population affected were children under 3 years of age. The predominant viruses detected were respiratory syncytial virus (RSV) and influenza A and B. The positive rate of viral testing was higher in children under 6 years of age compared to those aged 6-14 years. RSV infections were more common among children under 3 years old, while influenza A and B infections were more common in older children. Boys had a higher positive rate than girls. Virus infections peaked in the autumn and winter seasons and were lowest in spring and summer. The study found a significant decrease in the number of samples and positive rates of respiratory viruses compared to before the COVID-19 epidemic, possibly due to nonpharmaceutical interventions (NPIs) implemented during the pandemic.

Broadly, the authors demonstrate that the number of ARTIs decreased, however, the expected relative positivity proportions of ARTIs by species, age and seasonality

remained consistent during the COVID-19 pandemic implying that NPIs impacted non-COVID-19 respiratory viruses in a somewhat predictable way that can be applied in future pandemics or during respiratory seasons.

Major Concerns:

- The lack of vaccination status (COVID-19, Flu, etc.) impacts the generalizability and interpretation of the results. The authors note this as a limitation and also note China's relatively low Flu vaccination rates.

Thank you very much for your advice. From the results of this paper, we know that influenza A, influenza B and respiratory syncytial virus (RSV) are the three most common viruses in ARTIs. There is still no RSV vaccine for children. The impact of influenza vaccines is discussed in this article. The vaccination of COVID-19 vaccine in Chinese children began in October 2021, and the first round of vaccination was completed at the end of December 2021. Compared with the rates of positivity for respiratory viruses from January to February 2020, there was no significant change in 2021. For the above reasons, we have not discussed the impact of vaccines in this paper. However, your suggestion is very useful for us, and we added the corresponding content to the article.

Lines 271-277: “The vaccination of COVID-19 vaccine in Chinese children began in October 2021, and the first round of vaccination was completed at the end of December 2021. We analyzed the rates of positivity for respiratory viruses in 2022.1-2022.2. Compared with the positive proportion for respiratory viruses from January to February 2021, there was no significant change in 2022. The observation time after the new coronavirus vaccination in children was only 2 months, and a longer time is needed in future studies.”

- The impact of non-NPI pandemic-related factors is not discussed. I would advise a section of the discussion dedicated to addressing some of these factors, such as: reporting practices, variations in patterns of other infectious diseases/coinfections, changes in healthcare-seeking behavior, mistrust in science/healthcare, etc.

Thank you very much for your advice. We added the relevant content to the discussion.

Line 294-312: “Meanwhile, the impact of non-NPIs may have had an effect in reducing the rate of respiratory virus infection in children. Parents are not willing to take their sick children to the

hospital during the COVID-19 pandemic. They are worried about nosocomial infections, unwilling to be isolated from their children or reluctant to burden hospitals during the COVID-19 pandemic. Consequently, they may choose to delay treatment or seek medical help online. On the one hand, this reduces the spread of respiratory viruses; on the other hand, it also reduces the number of respiratory viruses detected^[36]. At the same time, due to isolation, less travel reduces air pollution, parents have more time to take care of sick children at home, and children may recover faster, thus indirectly reducing the probability of respiratory virus infection^[37]. The introduction of COVID-19 containment policies and public awareness campaigns may have reduced the transmission of *Streptococcus pneumoniae*, *Haemophilus influenzae*, and *Neisseria meningitidis*, thereby greatly reducing life-threatening invasive diseases in many countries worldwide^[38]. Studies have revealed that the number of admissions for respiratory infections in children caused mainly by bacteria has decreased significantly. The number of admissions for pneumonia decreased by 60%, that for otitis media decreased by 74%, and that for tonsillitis decreased by 66% after the outbreak of the epidemic. The number of bacterial and viral coinfections also decreased^[39].”

Minor Concerns:

Broad:

- Given the nature of a single center study, I would advise including details about the medical center and the patient population to give readers better context: geographic info, socioeconomic data, healthcare access, etc.

Thank you very much for your advice. We added the relevant content to the article.

Line 90-100:” The West China Second University Hospital of Sichuan University, which is located in Chengdu Sichuan province. The hospital was one of the first national top tertiary hospitals and serves as a medical center for women and children in southwestern China. It plays an important role in medical services, education, research, disease prevention, and healthcare. The hospital has 24 clinical departments/divisions and 7 medical supporting departments. As a referral center in Southwest China, we provide medical services to women and children with critical diseases. In 2022, we had 3.54 million outpatient and emergency visits, discharged 90,000 patients, performed 125,000 operations, and delivered 22,000 babies. Chengdu's resident population was 21.26 million, and the number of 0-14-year-old children was 2.78 million at the end of 2022.”

- Were comorbidities of patients analyzed? If so, I would include this data as it can impact ARTIs.

I am very sorry that we focused on the changes in the epidemic trend of respiratory viruses during the COVID-19 pandemic and did not analyze the comorbidities of patients in this article. We have added the relevant content to the limitations of this article. You have given us a very good advice, and we will pay attention to this field in future studies.

Lines 322-323:” Third, the comorbidities of patients were not analyzed in this study; we

will focus on this topic in future investigations.”

- Are there any available measures of adherence to NPIs as time went on in the 2-year study? As pandemic fatigue set in, were people more likely to not adhere therefore increasing ARTIs over the study period?

Yes, there are available measures of adherence to NPIs. First, the NPIs are dynamically adjusted, not always unchanged. For example, when the epidemic is serious, measures such as home isolation and banning social gatherings may be taken. After the epidemic has eased, people can enter and leave public places by wearing masks and maintaining hand hygiene measures. Second, the health management department has increased publicity to let people know about the new coronavirus and enhance their awareness of self-protection.

Abstract

- Define the 2 years of the pandemic as dates vary between countries

Thank you for your advice. We have added the relevant content to the article.

Line 19: ” during the two years of the pandemic (2020.3.1-2022.2.28).”

- Define the age of study 'children' even if typical <18 years old

Thank you for your advice. We have added the relevant content to the article.

Lines 14-15: “A total of 7092 children (<14 years) with ARTIs were included in this study.”

- "Positive rate" should be changed to "Positive proportion" unless the data were analyzed over time.

Thank you for your advice. We have changed "Positive rate" to "Positive proportion" throughout the article.

- How were positive test proportions calculated if only children with ARTIs were included?

If data is available on all tested children, meaning those who tested negative too, it should be presented, even if briefly, to give context.

I apologize for our inaccurate description. The positive test proportions were calculated in tested children with ARTIs. We have modified the related sentence in the text.

Line 19-21: ” The proportion of positive viral test results among patients with ARTIs < 6 years of age was higher than that among patients with ARTIs aged 6-14 years (17.3% vs. 5.7%, $P < 0.01$).”

- NPIs are one of many items that could have influenced any change in ARTIs during the pandemic. NPIs should be defined here so readers are clear this includes masking, distancing, etc.

Thank you for your advice. We added the relevant content to the article.

Line 32-33:” including banning social gatherings, wearing face masks, home isolation, and maintaining hand hygiene”

- "The types of epidemic viruses [changed]". I do not see any context that backs up this claim in the conclusion. This should be made clearer or removed.

Thank you for your advice. We removed this sentence from the article.

Introduction

- Line 42 needs a citation if not included in line 43.

Line 42 is included in line 43, and we have added citations in lines 43-44.

Line 43-44:” Acute respiratory tract infections (ARTIs) are the most common disease in children and adults^[1,2]”

- Line 52: for all patients or just children?

I apologize for our inaccurate description. We have modified related sentence in the text.

Line 53-54:” (ADV) and influenza virus, are the most common pathogens associated with ARTIs in children^[9]”

- Line 56: citation

We have added citations in line 58.

Line 58:” increase the risk of incident asthma^[10]”

- Line 68: Is the year 2020 accurate to the epidemic? Or is this intended for the first 'wave'?

Peaks occurred later than 2020 in China.

I apologize, as the first 'wave' is intended. We have modified the related sentence in the text.

Line 70-71:” By the end of March 2020, China had successfully controlled the first wave of the COVID-19 epidemic peak”

- Lines 79-85: It's confusing to connect these two ideas: 1) NPIs likely impacted ARTIs during the COVID pandemic and 2) understanding the prevalence of respiratory viruses during the pandemic could provide early warning of possible epidemic respiratory diseases. Are we attempting to understand the impact of NPIs on non-COVID respiratory viruses or a baseline level of viral infection prevalence's?

We were attempting to understand the impact of NPIs on non-COVID respiratory viruses, and we

have modified the related sentence in the text.

Lines 81-85: “Through the analysis of the positive proportion of seven respiratory viruses, common virus types, epidemic season and age of onset, we aimed to clarify the impact of NPIs on non-COVID respiratory viruses among children during the COVID-19 pandemic. This could prevent antibiotic abuse and guide the effective treatment and prevention of ARTIs.”

Methods

- Line 88: Why were these specific dates chosen as bookends? Even if obvious, please add.

We choose these specific dates to connect with our previous research. We have added the relevant content to the article.

Line 88-90: “To continue our previous research, we chose and retrospectively analyzed children with suspected ARTIs who visited pediatricians at West China Second University Hospital of Sichuan University from March 1, 2020, to February 28, 2022.”

- Please add the size and basic description of the study hospital for context.

We added the relevant content to the article.

Line 90-100: “The West China Second University Hospital of Sichuan University, which is located in Chengdu Sichuan province. The hospital was one of the first national top tertiary hospitals and serves as a medical center for women and children in southwestern China. It plays an important role in medical services, education, research, disease prevention, and healthcare. The hospital has 24 clinical departments/divisions and 7 medical supporting departments. As a referral center in Southwest China, we provide medical services to women and children with critical diseases. In 2022, we had 3.54 million outpatient and emergency visits, discharged 90,000 patients, performed 125,000 operations, and delivered 22,000 babies. Chengdu's resident population was 21.26 million, and the number of 0-14-year-old children was 2.78 million at the end of 2022.”

- Why was 14 years of age chosen?

o Throughout the manuscript the age inclusion criteria of <14 years of age should be emphasized as it is not a standard age cutoff.

Because the study hospital is a women's and children's hospital. The patients who underwent pediatrics were not more than 14 years old. If patients are older than 14 years old, they usually go to adult clinics in China.

Line 101-102 : “Patients younger than 14 years of age (as in China, the children visiting pediatric clinics are mainly aged 0-14 years.)”

- Were nasal swabs collected during the healthcare encounter patients were originally identified in or were they recruited and returned? This is critical considering that the

sensitivity of assays decreases as time from symptom onset increases.

Nasal swabs were collected during the healthcare encounter, and patients were originally identified.

- Were nasal swabs collected and tested for clinical purposes and done in the clinical microbiology lab or were these completed by research staff?

Nasal swabs were collected and tested for clinical purposes in the clinical microbiology laboratory.

- How many patients were excluded?

A total of 638 patients were excluded, and we added the relevant content to the article.

Lines 129-130:” There were 638 patients excluded.”

Discussion

- Line 215: Is this finding consistent with previous studies or is it unique to this one?

We apologize for our inaccurate description. We have modified this sentence.

Lines 231-234:” During the study period, the number of boys infected with ARTIs and the positive proportion of virus detection were significantly higher than those of girls, which is consistent with previous studies, and it is reported that boys may be more susceptible to ARTIs than girls.”

- Lines 218: The assumption of activity level differences between boys and girls should be removed unless data or relevant citations specific to the study area are added.

Thank you very much for your advice, we have removed this sentence in the text.

Reviewer #2

Reviewer Comments for Gao et al

The paper is overall a very interesting perspective on how strict restrictions implemented to curb the spread of Covid-19 infection had an effect on other respiratory pathogens. Gao et al took a retrospective look at samples obtained from March 2020 to March 2022, several months prior to the restriction being lifted in China. The paper is a simple yet effective look into how viral respiratory trends lifted, including a shift from influenza A dominance to RSV dominance, decrease in cases at the center where specimens were collected, and a remaining bias towards boys above the age of 3 having a higher likelihood for infection. The limitation of the study include and are not limited to being a single-centre study, samples only undergoing initial direct immunofluorescence assay testing and no PCR testing, not properly examining differences in viruses from inpatient vs outpatient patients, and the exclusion of findings for any co-infections.

There are some minor grammatical errors that should be addressed:

Line 53 - “since” instead of “because” line 55

Line 58 – “deserve” line 60

Line 82 – remove “us”

Thank you. We have revised those grammatical errors.

Other comments:

Abstract

Line 13 – study examined only viral respiratory specimens

Thank you for your advice. We added the relevant content to the text.

Line 13-14: “This study investigated respiratory viral specimens from children with ARTIs.”

Lines 14 and 19 – include the time frame in abstract

We have added the relevant content to the text.

Line 13:” (2020.3.1-2022.2.28)”

Line 19:” (2020.3.1-2022.2.28)”

Line 25 – peak virus infections occurred in the autumn and winter seasons in both years examined?

Yes, we have modified the related sentence in the text.

Line 27:” lowest activity level occurred in spring and summer of the two years.”

Materials and Methods

Line 98 – Any PCR testing performed? Any repeat testing?

The direct immunofluorescence assay is relatively simple and inexpensive and is widely used in clinical respiratory virus testing. Quality control will be performed in each test to ensure that the results are reliable. At the same time, if the sample test results were suspicious, they needed to be verified by PCR before inclusion in this study.

Results

Did the authors or clinicians look into coinfections?

Yes, in the stage of data collection and analysis, we exclude the ARITs combined with intestinal virus infection because this paper mainly analyzes and discusses respiratory virus infection. If you have any questions, please don't hesitate let me know.

Why did the clinicians not include metapneumovirus as part of their testing?

Because the direct immunofluorescence assay kit did not contain metapneumovirus, to ensure the consistency of the results, there was no metapneumovirus testing. The clinician tested for metapneumovirus by PCR.

Line 136 – Suggestion to authors to provide a figure to show the differences in age and the different viral infections? Table 1 should be mentioned again.

Thank you for your advice. We have added a figure as Supplement Figure 1 to show the differences in age and the different viral infections.

Lines 151-152:” (Supplement figure 1).”

Line 480-483:” Supplement figure1 the differences in age and the different viral infections. Flu A =influenza A; Flu B= influenza B; PIV I= parainfluenza virus I; PIV II= parainfluenza virus II; PIV III =parainfluenza virus III; ADV = adenovirus, RSV =

respiratory syncytial virus, D= Day, M=Month, Y=Year.”

Line 161 – authors mention the distribution of positive cases and viruses in boys versus girls during the pandemic. Please also mention distribution see prior to Covid pandemic.

Thank you for your advice. The distribution of positive cases before the COVID-19 pandemic has been mentioned and analyzed in our published article, and we discuss this content in lines 231-234. If you have any questions, please do not hesitate let me know.

Lines 132, 157 and 167 – The authors state the number of positive samples were 637 the first year and 992 the second year; these numbers do not match up to 1635 total positive samples reported.

I apologize that we wrote the numbers incorrectly when writing this article. The correct number is 643 instead of 637, which we have corrected in the full text. Thank you again.

Line 172:” The number of positive samples in the first year was 643”.

Discussion

The authors should elaborate more on the differences in viruses seen in outpatient vs. inpatient samples.

Thank you for your advice. We have added relevant content to the discussion.

Lines 238-244:” In this study, the number of inpatients was significantly higher than that of outpatients. The possible reason is that this testing requires a long time, and outpatients hope to obtain the test results as soon as possible to visit the doctor, resulting in a smaller number of tests than those for inpatients. At the same time, the main positive proportion of virus detection among outpatients involved RSV, influenza A and B, and the main positive proportion of virus detection among inpatients involved RSV, which is consistent with the relevant research reported [23, 24].”

Although significant, a change from 11813 samples precovid to 7092 samples reported during COVID is not as dramatic due to the lockdowns. Was this also due to less testing done prior to covid?

Thank you very much for your advice. There is a possibility that the COVID-19 pandemic has led to a reduction in the number of people being tested. We also added this part to the discussion in the article.

Lines 294-312: “Meanwhile, the impact of non-NPIs may have had an effect in reducing the rate of respiratory virus infection in children. Parents are not willing to take their sick children to the hospital during the COVID-19 pandemic. They are worried about nosocomial infections, unwilling to be isolated from their children or reluctant to burden hospitals during the COVID-19 pandemic. Consequently, they may choose to delay treatment or seek medical help online.

On the one hand, this reduces the spread of respiratory viruses; on the other hand, it also reduces the number of respiratory viruses detected^[36]. At the same time, due to isolation, less travel reduces air pollution, parents have more time to take care of sick children at home, and children may recover faster, thus indirectly reducing the probability of respiratory virus infection^[37]. The introduction of COVID-19 containment policies and public awareness campaigns may have reduced the transmission of *Streptococcus pneumoniae*, *Haemophilus influenzae*, and *Neisseria meningitidis*, thereby greatly reducing life-threatening invasive diseases in many countries worldwide^[38]. Studies have revealed that the number of admissions for respiratory infections in children caused mainly by bacteria has decreased significantly. The number of admissions for pneumonia decreased by 60%, that for otitis media decreased by 74%, and that for tonsillitis decreased by 66% after the outbreak of the epidemic. The number of bacterial and viral coinfections also decreased^[39].”

Line 201 – state percentage of decrease for children with respiratory infections and compare this percentage with the children in this study

Thank you for your advice. We have added relevant content to the discussion.

Lines 217-220: “A study showed that the number of children hospitalized for infectious diseases decreased by 37%, among whom the number of children with respiratory infections decreased by 33%^[20]. Similar reductions were observed in this study, with decreases of 31% for inpatients.”

Lines 206-208 is unclear and difficult to understand. “May not have had as great an effect” is unclear.

We apologize that we have modified this sentence in the article.

Lines 212-214: “On the other hand, parents' concerns about sick children and their seeking of timely clinical care may not play an important role”

Line 215 - is there also a genetic predisposition to boys having a higher rate of infections? Were the ratio of virally - infected boys vs. girls normalized to an uninfected population?

Thank you very much for your advice. The study revealed that the genetic susceptibility of children to respiratory diseases is related to the type of virus and specific respiratory diseases of patients. There are few studies indicating that the genetic susceptibility of children to respiratory infectious diseases is related to sex^[1-3]. Meanwhile, in the gender composition of children in Chengdu, boys accounted for 50.26%, girls accounted for 49.74%, and the proportion of boys and girls was basically the same. Therefore, we did not normalize the ratio of virally infected boys vs. girls to an uninfected population. If you have any questions, please do not hesitate let me know.

Line 220 – Authors should provide the citation in their own words.

Thank you for your advice. We have modified this sentence.

Line 235-238: “The possible reason is the waning of maternal antibodies after 6 months and the fact that the immune system gradually matures after 3 years of age. At this age stage, children lack antibodies against viral infections and are

more easily infected by viruses”

Line 224 – The paragraph can be reworded to better compare what was first previously seen and what is seen during the pandemic

Thank you for your advice. We have modified the relevant content in the text.

Line 245-257: “Our previous research revealed that children under 5 years of age were mainly infected with RSV and influenza A, and children aged 5-10 years were mainly infected with influenza A^[10]. During the pandemic, we found that children under 3 years of age were mainly infected with RSV, children aged 3-6 years were mainly infected with influenza A, and children aged 6-14 years were mainly infected with influenza A. Relevant studies have also shown that the population susceptible to RSV is mainly infants and young children, and the detection rate of RSV decreased with age, owing to the maturation of the immune system^[25, 26]. According to statistics, RSV causes at least 3.4 million hospitalizations and between 66,000 and 199,000 deaths among young children under 5 years of age every year^[27]. Meanwhile, the influenza virus prevalence increased with age, which was similar to previous reports^[28, 29]. Based on the results of this study, it is possible that there was no significant change in the population susceptible to viruses after the COVID-19 pandemic.”

Line 280 – Include % decrease

We have added the relevant content to the text.

Line 328:” (40% and 47%, respectively)”

Figures

Figure 1 Include 2018-2020 data if possible. Can be broken up to three different figures and further broken down by boys, girls, and total.

Thank you for your advice. We have modified the figure and added the relevant content to the text.

Line 154-165: “The seasonal distribution of the sample positive proportion during the study period is shown in Figure 1. The positive proportions of boys, girls and total in the autumn and winter seasons were significantly higher than those in the spring and summer seasons during the study period. In the first year, the positive proportion of boys was highest in January (24.4%) and lowest in July (2.2%) (Fig. 1A). The positive proportion of girls was highest in January (11.5%) and lowest in May (1.4%) (Fig. 1B). The total positive proportion was highest in January (36.0%) and lowest in July (3.9%) in this year (Fig. 1C). In the second year, the positive proportion of boys was highest in October (44.1%) and lowest in June (5.0%) (Fig. 1A). The positive proportion of girls was highest in October (30.5%) and lowest in April to June (3.2%-3.3%) (Fig. 1B). The total positive proportion was highest in October (74.6%) and lowest in June (8.3%) in this year (Fig. 1C). Meanwhile, the positive proportion of boys was higher than that of girls.”

Figures 2 and 3 Another figure can be included to further breakdown the different rates of infection in boys vs. girls

Thank you for your advice. We have modified the figure and added the relevant content to the text.

Lines 177-179:” The main positive proportion of virus detection among boys was RSV and influenza A, and the main positive proportion of virus detection among girls was RSV and influenza A (Fig. 2D).”

Lines 191-194:” The main positive proportion of virus detection among boys was RSV, influenza A and B, and the main positive proportion of virus detection among girls was RSV, influenza A and B (Fig. 3D).”

- [1] DRYSDALE S B, PRENDERGAST M, ALCAZAR M, et al. Genetic predisposition of RSV infection-related respiratory morbidity in preterm infants[J]. Eur J Pediatr, 2014, 173(7): 905-912.
- [2] DRYSDALE S B, MILNER A D, GREENOUGH A. Respiratory syncytial virus infection and chronic respiratory morbidity - is there a functional or genetic predisposition?[J]. Acta Paediatr, 2012, 101(11): 1114-1120.
- [3] DRYSDALE S B, ALCAZAR M, WILSON T, et al. Functional and genetic predisposition to rhinovirus lower respiratory tract infections in prematurely born infants[J]. Eur J Pediatr, 2016, 175(12): 1943-1949.

Re: Spectrum02614-23R1 (Epidemiological characteristics of respiratory viruses in children during the COVID-19 epidemic in Chengdu, China)

Dear Ms. leiwen peng:

Thank you for the privilege of reviewing your work. Below you will find my comments, instructions from the Spectrum editorial office, and the reviewer comments (in the attachment).

Revision Guidelines

Sincerely,
Tulip Jhaveri
Editor
Microbiology Spectrum

Reviewer #1 (Comments for the Author):

N/A

Reviewer #2 (Comments for the Author):

Thank you for all the clarifications and edits to make the paper stronger. Please see attached comments for minor additions.

Reviewer comments:

Reviewer #1 (Comments for the Author):

Summary of Key Findings:

This retrospective study aimed to analyze the prevalence of respiratory viruses in children with acute respiratory tract infections (ARTIs) during the COVID-19 pandemic. The study included 7,092 children with ARTIs, with a higher proportion of boys and a median age of 1 year and 5 months. The main population affected were children under 3 years of age. The predominant viruses detected were respiratory syncytial virus (RSV) and influenza A and B. The positive rate of viral testing was higher in children under 6 years of age compared to those aged 6-14 years. RSV infections were more common among children under 3 years old, while influenza A and B infections were more common in older children. Boys had a higher positive rate than girls. Virus infections peaked in the autumn and winter seasons and were lowest in spring and summer. The study found a significant decrease in the number of samples and positive rates of respiratory viruses compared to before the COVID-19 epidemic, possibly due to nonpharmaceutical interventions (NPIs) implemented during the pandemic.

Broadly, the authors demonstrate that the number of ARTIs decreased, however, the expected relative positivity proportions of ARTIs by species, age and seasonality

remained consistent during the COVID-19 pandemic implying that NPIs impacted non-COVID-19 respiratory viruses in a somewhat predictable way that can be applied in future pandemics or during respiratory seasons.

Major Concerns:

- The lack of vaccination status (COVID-19, Flu, etc.) impacts the generalizability and interpretation of the results. The authors note this as a limitation and also note China's relatively low Flu vaccination rates.

Thank you very much for your advice. From the results of this paper, we know that influenza A, influenza B and respiratory syncytial virus (RSV) are the three most common viruses in ARTIs. There is still no RSV vaccine for children. The impact of influenza vaccines is discussed in this article. The vaccination of COVID-19 vaccine in Chinese children began in October 2021, and the first round of vaccination was completed at the end of December 2021. Compared with the rates of positivity for respiratory viruses from January to February 2020, there was no significant change in 2021. For the above reasons, we have not discussed the impact of vaccines in this paper. However, your suggestion is very useful for us, and we added the corresponding content to the article.

Lines 271-277: “The vaccination of COVID-19 vaccine in Chinese children began in October 2021, and the first round of vaccination was completed at the end of December 2021. We analyzed the rates of positivity for respiratory viruses in 2022.1-2022.2. Compared with the positive proportion for respiratory viruses from January to February 2021, there was no significant change in 2022. The observation time after the new coronavirus vaccination in children was only 2 months, and a longer time is needed in future studies.”

- The impact of non-NPI pandemic-related factors is not discussed. I would advise a section of the discussion dedicated to addressing some of these factors, such as: reporting practices, variations in patterns of other infectious diseases/coinfections, changes in healthcare-seeking behavior, mistrust in science/healthcare, etc.

Thank you very much for your advice. We added the relevant content to the discussion.

Line 294-312: “Meanwhile, the impact of non-NPIs may have had an effect in reducing the rate of respiratory virus infection in children. Parents are not willing to take their sick children to the

hospital during the COVID-19 pandemic. They are worried about nosocomial infections, unwilling to be isolated from their children or reluctant to burden hospitals during the COVID-19 pandemic. Consequently, they may choose to delay treatment or seek medical help online. On the one hand, this reduces the spread of respiratory viruses; on the other hand, it also reduces the number of respiratory viruses detected^[36]. At the same time, due to isolation, less travel reduces air pollution, parents have more time to take care of sick children at home, and children may recover faster, thus indirectly reducing the probability of respiratory virus infection^[37]. The introduction of COVID-19 containment policies and public awareness campaigns may have reduced the transmission of *Streptococcus pneumoniae*, *Haemophilus influenzae*, and *Neisseria meningitidis*, thereby greatly reducing life-threatening invasive diseases in many countries worldwide^[38]. Studies have revealed that the number of admissions for respiratory infections in children caused mainly by bacteria has decreased significantly. The number of admissions for pneumonia decreased by 60%, that for otitis media decreased by 74%, and that for tonsillitis decreased by 66% after the outbreak of the epidemic. The number of bacterial and viral coinfections also decreased^[39].”

Minor Concerns:

Broad:

- Given the nature of a single center study, I would advise including details about the medical center and the patient population to give readers better context: geographic info, socioeconomic data, healthcare access, etc.

Thank you very much for your advice. We added the relevant content to the article.

Line 90-100:” The West China Second University Hospital of Sichuan University, which is located in Chengdu Sichuan province. The hospital was one of the first national top tertiary hospitals and serves as a medical center for women and children in southwestern China. It plays an important role in medical services, education, research, disease prevention, and healthcare. The hospital has 24 clinical departments/divisions and 7 medical supporting departments. As a referral center in Southwest China, we provide medical services to women and children with critical diseases. In 2022, we had 3.54 million outpatient and emergency visits, discharged 90,000 patients, performed 125,000 operations, and delivered 22,000 babies. Chengdu's resident population was 21.26 million, and the number of 0-14-year-old children was 2.78 million at the end of 2022.”

- Were comorbidities of patients analyzed? If so, I would include this data as it can impact ARTIs.

I am very sorry that we focused on the changes in the epidemic trend of respiratory viruses during the COVID-19 pandemic and did not analyze the comorbidities of patients in this article. We have added the relevant content to the limitations of this article. You have given us a very good advice, and we will pay attention to this field in future studies.

Lines 322-323:” Third, the comorbidities of patients were not analyzed in this study; we

will focus on this topic in future investigations.”

- Are there any available measures of adherence to NPIs as time went on in the 2-year study? As pandemic fatigue set in, were people more likely to not adhere therefore increasing ARTIs over the study period?

Yes, there are available measures of adherence to NPIs. First, the NPIs are dynamically adjusted, not always unchanged. For example, when the epidemic is serious, measures such as home isolation and banning social gatherings may be taken. After the epidemic has eased, people can enter and leave public places by wearing masks and maintaining hand hygiene measures. Second, the health management department has increased publicity to let people know about the new coronavirus and enhance their awareness of self-protection.

Abstract

- Define the 2 years of the pandemic as dates vary between countries

Thank you for your advice. We have added the relevant content to the article.

Line 19: ” during the two years of the pandemic (2020.3.1-2022.2.28).”

- Define the age of study 'children' even if typical <18 years old

Thank you for your advice. We have added the relevant content to the article.

Lines 14-15: “A total of 7092 children (<14 years) with ARTIs were included in this study.”

- "Positive rate" should be changed to "Positive proportion" unless the data were analyzed over time.

Thank you for your advice. We have changed "Positive rate" to "Positive proportion" throughout the article.

- How were positive test proportions calculated if only children with ARTIs were included?

If data is available on all tested children, meaning those who tested negative too, it should be presented, even if briefly, to give context.

I apologize for our inaccurate description. The positive test proportions were calculated in tested children with ARTIs. We have modified the related sentence in the text.

Line 19-21: ” The proportion of positive viral test results among patients with ARTIs < 6 years of age was higher than that among patients with ARTIs aged 6-14 years (17.3% vs. 5.7%, $P < 0.01$).”

- NPIs are one of many items that could have influenced any change in ARTIs during the pandemic. NPIs should be defined here so readers are clear this includes masking, distancing, etc.

Thank you for your advice. We added the relevant content to the article.

Line 32-33:” including banning social gatherings, wearing face masks, home isolation, and maintaining hand hygiene”

- "The types of epidemic viruses [changed]". I do not see any context that backs up this claim in the conclusion. This should be made clearer or removed.

Thank you for your advice. We removed this sentence from the article.

Introduction

- Line 42 needs a citation if not included in line 43.

Line 42 is included in line 43, and we have added citations in lines 43-44.

Line 43-44:” Acute respiratory tract infections (ARTIs) are the most common disease in children and adults^[1,2]”

- Line 52: for all patients or just children?

I apologize for our inaccurate description. We have modified related sentence in the text.

Line 53-54:” (ADV) and influenza virus, are the most common pathogens associated with ARTIs in children^[9]”

- Line 56: citation

We have added citations in line 58.

Line 58:” increase the risk of incident asthma^[10]”

- Line 68: Is the year 2020 accurate to the epidemic? Or is this intended for the first 'wave'?

Peaks occurred later than 2020 in China.

I apologize, as the first 'wave' is intended. We have modified the related sentence in the text.

Line 70-71:” By the end of March 2020, China had successfully controlled the first wave of the COVID-19 epidemic peak”

- Lines 79-85: It's confusing to connect these two ideas: 1) NPIs likely impacted ARTIs during the COVID pandemic and 2) understanding the prevalence of respiratory viruses during the pandemic could provide early warning of possible epidemic respiratory diseases. Are we attempting to understand the impact of NPIs on non-COVID respiratory viruses or a baseline level of viral infection prevalence's?

We were attempting to understand the impact of NPIs on non-COVID respiratory viruses, and we

have modified the related sentence in the text.

Lines 81-85: “Through the analysis of the positive proportion of seven respiratory viruses, common virus types, epidemic season and age of onset, we aimed to clarify the impact of NPIs on non-COVID respiratory viruses among children during the COVID-19 pandemic. This could prevent antibiotic abuse and guide the effective treatment and prevention of ARTIs.”

Methods

- Line 88: Why were these specific dates chosen as bookends? Even if obvious, please add.

We choose these specific dates to connect with our previous research. We have added the relevant content to the article.

Line 88-90: “To continue our previous research, we chose and retrospectively analyzed children with suspected ARTIs who visited pediatricians at West China Second University Hospital of Sichuan University from March 1, 2020, to February 28, 2022.”

- Please add the size and basic description of the study hospital for context.

We added the relevant content to the article.

Line 90-100: “The West China Second University Hospital of Sichuan University, which is located in Chengdu Sichuan province. The hospital was one of the first national top tertiary hospitals and serves as a medical center for women and children in southwestern China. It plays an important role in medical services, education, research, disease prevention, and healthcare. The hospital has 24 clinical departments/divisions and 7 medical supporting departments. As a referral center in Southwest China, we provide medical services to women and children with critical diseases. In 2022, we had 3.54 million outpatient and emergency visits, discharged 90,000 patients, performed 125,000 operations, and delivered 22,000 babies. Chengdu's resident population was 21.26 million, and the number of 0-14-year-old children was 2.78 million at the end of 2022.”

- Why was 14 years of age chosen?

o Throughout the manuscript the age inclusion criteria of <14 years of age should be emphasized as it is not a standard age cutoff.

Because the study hospital is a women's and children's hospital. The patients who underwent pediatrics were not more than 14 years old. If patients are older than 14 years old, they usually go to adult clinics in China.

Line 101-102 : “Patients younger than 14 years of age (as in China, the children visiting pediatric clinics are mainly aged 0-14 years.)”

- Were nasal swabs collected during the healthcare encounter patients were originally identified in or were they recruited and returned? This is critical considering that the

sensitivity of assays decreases as time from symptom onset increases.

Nasal swabs were collected during the healthcare encounter, and patients were originally identified.

- Were nasal swabs collected and tested for clinical purposes and done in the clinical microbiology lab or were these completed by research staff?

Nasal swabs were collected and tested for clinical purposes in the clinical microbiology laboratory.

- How many patients were excluded?

A total of 638 patients were excluded, and we added the relevant content to the article.

Lines 129-130:” There were 638 patients excluded.”

Discussion

- Line 215: Is this finding consistent with previous studies or is it unique to this one?

We apologize for our inaccurate description. We have modified this sentence.

Lines 231-234:” During the study period, the number of boys infected with ARTIs and the positive proportion of virus detection were significantly higher than those of girls, which is consistent with previous studies, and it is reported that boys may be more susceptible to ARTIs than girls.”

- Lines 218: The assumption of activity level differences between boys and girls should be removed unless data or relevant citations specific to the study area are added.

Thank you very much for your advice, we have removed this sentence in the text.

Reviewer #2

Reviewer Comments for Gao et al

The paper is overall a very interesting perspective on how strict restrictions implemented to curb the spread of Covid-19 infection had an effect on other respiratory pathogens. Gao et al took a retrospective look at samples obtained from March 2020 to March 2022, several months prior to the restriction being lifted in China. The paper is a simple yet effective look into how viral respiratory trends lifted, including a shift from influenza A dominance to RSV dominance, decrease in cases at the center where specimens were collected, and a remaining bias towards boys above the age of 3 having a higher likelihood for infection. The limitation of the study include and are not limited to being a single-centre study, samples only undergoing initial direct immunofluorescence assay testing and no PCR testing, not properly examining differences in viruses from inpatient vs outpatient patients, and the exclusion of findings for any co-infections.

There are some minor grammatical errors that should be addressed:

Line 53 - “since” instead of “because” line 55

Line 58 – “deserve” line 60

Line 82 – remove “us”

Thank you. We have revised those grammatical errors.

Other comments:

Abstract

Line 13 – study examined only viral respiratory specimens

Thank you for your advice. We added the relevant content to the text.

Line 13-14: “This study investigated respiratory viral specimens from children with ARTIs.”

Lines 14 and 19 – include the time frame in abstract

We have added the relevant content to the text.

Line 13:” (2020.3.1-2022.2.28)”

Line 19:” (2020.3.1-2022.2.28)”

Line 25 – peak virus infections occurred in the autumn and winter seasons in both years examined?

Yes, we have modified the related sentence in the text.

Line 27:” lowest activity level occurred in spring and summer of the two years.”

Materials and Methods

Line 98 – Any PCR testing performed? Any repeat testing?

The direct immunofluorescence assay is relatively simple and inexpensive and is widely used in clinical respiratory virus testing. Quality control will be performed in each test to ensure that the results are reliable. At the same time, if the sample test results were suspicious, they needed to be verified by PCR before inclusion in this study. **Please include this information in the materials and methods section.**

Results

Did the authors or clinicians look into coinfections?

Yes, in the stage of data collection and analysis, we exclude the ARITs combined with intestinal virus infection because this paper mainly analyzes and discusses respiratory virus infection. If you have any questions, please don't hesitate let me know.

Why did the clinicians not include metapneumovirus as part of their testing?

Because the direct immunofluorescence assay kit did not contain metapneumovirus, to ensure the consistency of the results, there was no metapneumovirus testing. The clinician tested for metapneumovirus by PCR. **Please include this information in the paper**

Line 136 – Suggestion to authors to provide a figure to show the differences in age and the different viral infections? Table 1 should be mentioned again.

Thank you for your advice. We have added a figure as Supplement Figure 1 to show the differences in age and the different viral infections.

Lines 151-152:” (Supplement figure 1).”

Line 480-483:” Supplement figure1 the differences in age and the different viral infections. Flu A =influenza A; Flu B= influenza B; PIV I= parainfluenza virus I; PIV

II= parainfluenza virus II; PIV III =parainfluenza virus III; ADV = adenovirus, RSV = respiratory syncytial virus, D= Day, M=Month, Y=Year.”

Line 161 – authors mention the distribution of positive cases and viruses in boys versus girls during the pandemic. Please also mention distribution see prior to Covid pandemic.

Thank you for your advice. The distribution of positive cases before the COVID-19 pandemic has been mentioned and analyzed in our published article, and we discuss this content in lines 231-234. If you have any questions, please do not hesitate let me know.

Lines 132, 157 and 167 – The authors state the number of positive samples were 637 the first year and 992 the second year; these numbers do not match up to 1635 total positive samples reported.

I apologize that we wrote the numbers incorrectly when writing this article. The correct number is 643 instead of 637, which we have corrected in the full text. Thank you again.

Line 172:” The number of positive samples in the first year was 643”.

Discussion

The authors should elaborate more on the differences in viruses seen in outpatient vs. inpatient samples.

Thank you for your advice. We have added relevant content to the discussion.

Lines 238-244:” In this study, the number of inpatients was significantly higher than that of outpatients. The possible reason is that this testing requires a long time, and outpatients hope to obtain the test results as soon as possible to visit the doctor, resulting in a smaller number of tests than those for inpatients. At the same time, the main positive proportion of virus detection among outpatients involved RSV, influenza A and B, and the main positive proportion of virus detection among inpatients involved RSV, which is consistent with the relevant research reported ^[23, 24] .”

Although significant, a change from 11813 samples precovid to 7092 samples reported during COVID is not as dramatic due to the lockdowns. Was this also due to less testing done prior to covid?

Thank you very much for your advice. There is a possibility that the COVID-19 pandemic has led to a reduction in the number of people being tested. We also added this part to the discussion in the article.

Lines 294-312: “Meanwhile, the impact of non-NPIs may have had an effect in reducing the rate of respiratory virus infection in children. Parents are not willing to take their sick children to the hospital during the COVID-19 pandemic. They are worried about nosocomial infections, unwilling to be isolated from their children or reluctant to burden hospitals during the COVID-19 pandemic.

Consequently, they may choose to delay treatment or seek medical help online. On the one hand, this reduces the spread of respiratory viruses; on the other hand, it also reduces the number of respiratory viruses detected^[36]. At the same time, due to isolation, less travel reduces air pollution, parents have more time to take care of sick children at home, and children may recover faster, thus indirectly reducing the probability of respiratory virus infection^[37]. The introduction of COVID-19 containment policies and public awareness campaigns may have reduced the transmission of *Streptococcus pneumoniae*, *Haemophilus influenzae*, and *Neisseria meningitidis*, thereby greatly reducing life-threatening invasive diseases in many countries worldwide^[38]. Studies have revealed that the number of admissions for respiratory infections in children caused mainly by bacteria has decreased significantly. The number of admissions for pneumonia decreased by 60%, that for otitis media decreased by 74%, and that for tonsillitis decreased by 66% after the outbreak of the epidemic. The number of bacterial and viral coinfections also decreased^[39].”

Line 201 – state percentage of decrease for children with respiratory infections and compare this percentage with the children in this study

Thank you for your advice. We have added relevant content to the discussion.

Lines 217-220: “A study showed that the number of children hospitalized for infectious diseases decreased by 37%, among whom the number of children with respiratory infections decreased by 33%^[20]. Similar reductions were observed in this study, with decreases of 31% for inpatients.”

Lines 206-208 is unclear and difficult to understand. “May not have had as great an effect” is unclear.

We apologize that we have modified this sentence in the article.

Lines 212-214: “On the other hand, parents' concerns about sick children and their seeking of timely clinical care may not play an important role”

Line 215 - is there also a genetic predisposition to boys having a higher rate of infections? Were the ratio of virally - infected boys vs. girls normalized to an uninfected population?

Thank you very much for your advice. The study revealed that the genetic susceptibility of children to respiratory diseases is related to the type of virus and specific respiratory diseases of patients. There are few studies indicating that the genetic susceptibility of children to respiratory infectious diseases is related to sex^[1-3]. Meanwhile, in the gender composition of children in Chengdu, boys accounted for 50.26%, girls accounted for 49.74%, and the proportion of boys and girls was basically the same. Therefore, we did not normalize the ratio of virally infected boys vs. girls to an uninfected population. If you have any questions, please do not hesitate let me know. **Please include this information in the discussion**

Line 220 – Authors should provide the citation in their own words.

Thank you for your advice. We have modified this sentence.

Line 235-238: “The possible reason is the waning of maternal antibodies after

6 months and the fact that the immune system gradually matures after 3 years of age. At this age stage, children lack antibodies against viral infections and are more easily infected by viruses”

Line 224 – The paragraph can be reworded to better compare what was first previously seen and what is seen during the pandemic

Thank you for your advice. We have modified the relevant content in the text.

Line 245-257: “Our previous research revealed that children under 5 years of age were mainly infected with RSV and influenza A, and children aged 5-10 years were mainly infected with influenza A^[10]. During the pandemic, we found that children under 3 years of age were mainly infected with RSV, children aged 3-6 years were mainly infected with influenza A, and children aged 6-14 years were mainly infected with influenza A. Relevant studies have also shown that the population susceptible to RSV is mainly infants and young children, and the detection rate of RSV decreased with age, owing to the maturation of the immune system^[25, 26]. According to statistics, RSV causes at least 3.4 million hospitalizations and between 66,000 and 199,000 deaths among young children under 5 years of age every year^[27]. Meanwhile, the influenza virus prevalence increased with age, which was similar to previous reports^[28, 29]. Based on the results of this study, it is possible that there was no significant change in the population susceptible to viruses after the COVID-19 pandemic.”

Line 280 – Include % decrease

We have added the relevant content to the text.

Line 328:” (40% and 47%, respectively)”

Figures

Figure 1 Include 2018-2020 data if possible. Can be broken up to three different figures and further broken down by boys, girls, and total.

Thank you for your advice. We have modified the figure and added the relevant content to the text.

Line 154-165: “The seasonal distribution of the sample positive proportion during the study period is shown in Figure 1. The positive proportions of boys, girls and total in the autumn and winter seasons were significantly higher than those in the spring and summer seasons during the study period. In the first year, the positive proportion of boys was highest in January (24.4%) and lowest in July (2.2%) (Fig. 1A). The positive proportion of girls was highest in January (11.5%) and lowest in May (1.4%) (Fig. 1B). The total positive proportion was highest in January (36.0%) and lowest in July (3.9%) in this year (Fig. 1C). In the second year, the positive proportion of boys was highest in October (44.1%) and lowest in June (5.0%) (Fig. 1A). The positive proportion of girls was highest in October (30.5%) and lowest in April to June (3.2%-3.3%) (Fig. 1B). The total positive proportion was highest in October (74.6%) and lowest in June (8.3%) in this year (Fig. 1C). Meanwhile, the positive proportion of boys was higher than that of girls.”

Figures 2 and 3 Another figure can be included to further breakdown the different rates of infection in boys vs. girls

Thank you for your advice. We have modified the figure and added the relevant content to the text.

Lines 177-179:” The main positive proportion of virus detection among boys was RSV and influenza A, and the main positive proportion of virus detection among girls was RSV and influenza A (Fig. 2D).”

Lines 191-194:” The main positive proportion of virus detection among boys was RSV, influenza A and B, and the main positive proportion of virus detection among girls was RSV, influenza A and B (Fig. 3D).”

- [1] DRYSDALE S B, PRENDERGAST M, ALCAZAR M, et al. Genetic predisposition of RSV infection-related respiratory morbidity in preterm infants[J]. Eur J Pediatr, 2014, 173(7): 905-912.
- [2] DRYSDALE S B, MILNER A D, GREENOUGH A. Respiratory syncytial virus infection and chronic respiratory morbidity - is there a functional or genetic predisposition?[J]. Acta Paediatr, 2012, 101(11): 1114-1120.
- [3] DRYSDALE S B, ALCAZAR M, WILSON T, et al. Functional and genetic predisposition to rhinovirus lower respiratory tract infections in prematurely born infants[J]. Eur J Pediatr, 2016, 175(12): 1943-1949.

Reviewer #2

Materials and Methods

Line 98 – Any PCR testing performed? Any repeat testing?

The direct immunofluorescence assay is relatively simple and inexpensive and is widely used in clinical respiratory virus testing. Quality control will be performed in each test to ensure that the results are reliable. At the same time, if the sample test results were suspicious, they needed to be verified by PCR before inclusion in this study. **Please include this information in the materials and methods section.**

Thank you for your advice. We have added this information in the materials and methods section.

Lines 114-118:” The direct immunofluorescence assay is relatively simple and inexpensive and is widely used in clinical respiratory virus testing. Quality control will be performed in each test to ensure that the results are reliable. At the same time, if the sample test results were suspicious, they needed to be verified by PCR before inclusion in this study.”

Results

Why did the clinicians not include metapneumovirus as part of their testing?

Because the direct immunofluorescence assay kit did not contain metapneumovirus, to ensure the consistency of the results, there was no metapneumovirus testing. The clinician tested for metapneumovirus by PCR.

Please include this information in the paper

Thank you for your advice. We have added this information in the discussion.

Lines 118-121:” The direct immunofluorescence assay kit did not contain metapneumovirus, to ensure the consistency of the results, there was no metapneumovirus testing. The clinician tested for metapneumovirus by PCR.”

Discussion

Line 215 – is there also a genetic predisposition to boys having a higher rate of infections? Were the ratio of virally-infected boys vs. girls normalized to an

uninfected population?

Thank you very much for your advice. The study revealed that the genetic susceptibility of children to respiratory diseases is related to the type of virus and specific respiratory diseases of patients. There are few studies indicating that the genetic susceptibility of children to respiratory infectious diseases is related to sex^[1-3]. Meanwhile, in the gender composition of children in Chengdu, boys accounted for 50.26%, girls accounted for 49.74%, and the proportion of boys and girls was basically the same. Therefore, we did not normalize the ratio of virally infected boys vs. girls to an uninfected population. If you have any questions, please do not hesitate let me know. **Please include this information in the discussion**

Thank you for your advice. We have added this information in the discussion.

Line 241-247: “The study revealed that the genetic susceptibility of children to respiratory diseases is related to the type of virus and specific respiratory diseases of patients. There are few studies indicating that the genetic susceptibility of children to respiratory infectious diseases is related to sex^[22-24]. Meanwhile, in the gender composition of children in Chengdu, boys accounted for 50.26%, girls accounted for 49.74%, and the proportion of boys and girls was basically the same. Therefore, we did not normalize the ratio of virally infected boys vs. girls to an uninfected population.”

Re: Spectrum02614-23R2 (Epidemiological characteristics of respiratory viruses in children during the COVID-19 epidemic in Chengdu, China)

Dear Ms. leiwen peng:

Your manuscript has been accepted, and I am forwarding it to the ASM production staff for publication. Your paper will first be checked to make sure all elements meet the technical requirements. ASM staff will contact you if anything needs to be revised before copyediting and production can begin. Otherwise, you will be notified when your proofs are ready to be viewed.

Sincerely,
Tulip Jhaveri
Editor
Microbiology Spectrum